# The relationship between new PCR positive cases and going out in public during the COVID-19 epidemic in Japan

Hiromichi Takahashi, Iori Terada, Takuya Higuchi[ORCID], Daisuke Takada, Jung-ho Shin, Susumu Kunisawa, Yuichi Imanaka[ORCID]*

Department of Healthcare Economics and Quality Management, Graduate School of Medicine, Kyoto University, Kyoto, Japan

* imanaka-y@umin.net

## Abstract

The suppression of the first wave of COVID-19 in Japan is assumedly attributed to people's increased risk perception after acquiring information from the government and media reports. In this study, going out in public amidst the spread of COVID-19 infections was investigated by examining new polymerase chain reaction (PCR) positive cases of COVID-19 and its relationship to four indicators of people going out in public (the people flow, the index of web searches for going outside, the number of times people browse restaurants, and the number of hotel guests, from the Regional Economic and Social Analysis System (V-RESAS). Two waves of COVID-19 infections were examined using cross-correlation analysis. In the first wave, all four indicators of going out changed to be opposite the change in new PCR positive cases, showing a lag period of –1 to +6 weeks. In the second wave, the same relationship was only observed for the index of web searches for going outside, and two indicators showed the positive lag period of +6 to +12 weeks after the change in new PCR positive cases. Moreover, each indicator in the second wave changed differently compared to the first wave. The complexity of people's behaviors around going out increased in the second wave, when policies and campaigns were implemented and people's attitudes were thought to have changed. In conclusion, the results suggest that policies may have influenced people's mobility, rather than the number of new PCR positive cases.

## 1. Introduction

The novel coronavirus disease 2019 (COVID-19)—caused by infection with severe acute respiratory syndrome coronavirus two (SARS-CoV-2), which can lead to severe pneumonia in infected humans—grew into an unprecedented global pandemic in early 2020. After the first outbreak in China in December 2019, the disease has continued to spread throughout the world, significantly impacting everyday life [1, 2]. In Japan, the first COVID-19 case was confirmed on January 16 [3], and since then, the number of cases and deaths has continuously fluctuated. The first wave of COVID-19 infections ended in May 2020 [4].

**Funding:** This work was supported by JSPS KAKENHI (Grant Number JP19H01075) from the Japan Society for the Promotion of Science (https://www.jsps.go.jp/english/e-grants/), and by the GAP Fund Program of Kyoto University, GAP Fund Program Type B (http://www.venture.saci.kyoto-u.ac.jp/?page_id=83#gp) to Y. I. The funders played no role in the study design, data collection and, data management, analysis, decision to publish, preparation, review and approval of the manuscript.

**Competing interests:** The authors have declared that no competing interests exist.

Several studies have demonstrated a link between people reducing their mobility and staying home and a reduction in the number of infections and deaths [5–10]. The government implemented various policies to influence and control people flow to reduce the spread of infection. For instance, the government designated COVID-19 as the equivalent of a category two infectious disease [11, 12] and postponed the Olympic games [13]. The government also requested various events be canceled [14] and that schools and high-risk facilities, such as bars, be closed [15, 16]. A state of emergency was declared [17–21], and people were asked to refrain from going out in public to reduce person-to-person transmission by 70–80% [22]. These measures were requested of the public but were not compulsory, and there were no penalties for disregarding these initiatives. Nevertheless, people followed the requests and refrained from going out in public to a certain extent.

When the first wave was contained, the government proposed a new public lifestyle to prevent the spread of infection. The guidance stipulated avoiding the "three Cs": closed spaces, crowded places, and close-contact settings [23]. Furthermore, the government and each prefecture monitored several infection indicators and set specific standards to combat subsequent infections [24–26]. These measures were intended to minimize the risk of repeating the negative impacts of the first wave from a long-term perspective while acknowledging the need to coexist with the virus. As an economic measure, the government provided 100,000 yen per person during the first wave [27]. However, the damage to the economy was tremendous [28] due to people's self-restraint in avoiding going out during the first wave of the pandemic. Several stimulus measures were implemented after the first wave, including subsidies for the travel and restaurant industries [29–34]. These economic stimulus measures were designed to encourage people to go out and move around while the infection was spreading, and it could be inferred that this increased the number of people traveling and eating out.

As previous studies have shown, the end of the first wave was achieved by people's self-restraint from going out in public, which was attributed to increased public awareness and understanding of the risks of the current situation through media reports [35–40]. In addition, the characteristics of Japanese culture and customs—greeting without shaking hands, hugging or kissing, and wearing cloth or paper facemasks to prevent respiratory infections and pollen allergies—may have contributed to the lower number of new polymerase chain reaction (PCR) positive cases and deaths per population compared to other countries [41]. Watanabe and Mizuno attributed the decrease in the number of people going out in public during the first wave to government announcements, such as daily news releases of new PCR positive cases [42, 43]. Meanwhile, anxiety and precautionary behaviors decreased as people became more aware of their subjective risk perception as an immediate threat [44]. Daily news reports, including new PCR positive cases, informed the public of the seriousness of the COVID-19 epidemic. Based on the available information of the situation, and including both objective and subjective reasoning, people then decide how they will behave, for example, choosing whether to go out in public or stay home. The information reported in the media is expected to influence individual behavior and either discourage or encourage going out, especially concerning the three Cs considered high risk. Previous studies showed the association between an increase in the number of new PCR positive cases and a decrease in people's mobility [45–50], and that the government declaration is effective to lead people to stay home [51]. In addition, studies in various countries showed that the degree of compliance with restrictions diminished over time [52–54].

The association between the new PCR positive cases and going out in public, the differences in people going out between the first and subsequent waves, and people's behavior among the different kinds of going out have not been clarified, especially in Japan. Thus, this study investigated the relationship between the new PCR positive cases and four indicators of going out,

namely the people flow; the index of web searches for going outside; the number of times people browse restaurants, and the number of hotel guests, by examining the lag of going out behaviors from the spread of infection. By examining the changes in people going out in public between the first and subsequent waves, and the underlying changes in people's awareness of the crisis as well as the effects of government policies, this study will provide insight for implementing effective countermeasures against the spread of infection.

## 2. Materials and methods

### 2-1. Data collection

**2-1-1. New PCR positive cases.** New PCR positive cases of COVID-19 are announced daily by the Ministry of Health, Labor, and Welfare. This number is based on the date each prefecture receives a positive case report from medical institutions, not necessarily the date of infection onset. The data was obtained from "Toyo Keizai Online", a Japanese publication that independently compiles data from the Ministry of Health, Labor, and Welfare [55]. The published data is an indicator for implementing various policies and informing media news that people see daily, thereby influencing public awareness. The number of cases is tallied weekly, Monday through Sunday.

**2-1-2. Four indicators of going out in public.** The Regional Economic and Social Analysis System (V-RESAS) website reports four public movement indicators based on data provided by the Office for Promotion of Regional Revitalization, the Cabinet Office, Government of Japan [56]. Data from multiple sources is compiled and V-RESAS shows the cumulative outcomes, showing weekly change rates compared to data covering the same period of the previous year [56].

The people flow shows the weekly change in the rate of people moving through several locations in Japan compared to the same week of the previous year. Agoop is a company that collects location information, as derived from their own applications and other companies that allow Agoop to use their location data [56, 57]. Agoop's data is used by the V-RESAS and various media outlets, including the Japanese public broadcaster NHK [58].

The index of web searches for going outside was derived from the data of Yahoo Japan Corporation, which uses AI technology to categorize words entered into Yahoo! Search [56, 59].

The number of times people browse restaurants was derived from restaurant information views on the Food Data Platform provided by Retty, a large-scale food business platform with 40 million monthly users [56, 60].

Accommodation data from the Tourism Forecast Platform, for which the Japan Travel and Tourism Association serves as the secretariat, provides the number of guests staying at hotels. The anonymized data was collected from travel agency storefronts and reservation sites, and as of September 2020 included more than 130 million stays [56, 61].

### 2-2. Definitions of COVID-19 epidemic periods

In this study, COVID-19 infections in Japan were divided into two periods. The first wave dated from January 16, 2020, when the first infected person was identified, and lasted until the end of May 2020 [4]. The second wave lasted from June 1, 2020, to the first week of November 2020.

### 2-3. Analysis

Cross-correlation analysis was performed to investigate the relationship between new PCR positive cases and each of the four indicators of going out in public for each of the first and

second waves. The cross-correlation function (CCF) describes the relationship between two time-series datasets, X(t) and Y(t), and has been used as a method to estimate the time lag between the datasets. One infectious disease study previously used this method for an outbreak of acute exanthematous illness in Brazil during 2014–2015, which was attributed to Zika virus, Guillain-Barre syndrome, and microcephaly [62]. In addition, CCF was used to estimate the speed of influenza epidemics by comparing the lag in drug sales between geographically separated pharmacies in Japan [63]. Moreover, it has been used to detect the correlation pattern between output and nominal variables in economics [64]. Following the previous study [63], CCF was defined as the following equation.

$$CCF(l) = \frac{E[X(t)Y(t+l)]}{\sqrt{E[X(t)^2]E[Y(t)^2]}} \qquad (1)$$

where *CCF(l)* denotes the cross-correlation function between two time series, *X(t)* and *Y(t)*; *E* [.] denotes the expectation over time *l* of the random variables inside the square brackets, and *l* denotes the lag.

Interpreting the results of CCF analysis focuses on two points: the sign of the lag and the sign of cross-correlation. For the former, if the lag is negative, the time series data *X(t)* is ahead of *Y(t)*. On the other hand, if the lag is positive, the time series data *X(t)* can be expressed as lagging behind *Y(t)*, i.e., the lag is judged by whether it is to the right side (i.e., the positive region) or the left side (i.e., the negative region). For the latter, when the cross-correlation is positive, the forms of the two indicators are similar; when it is negative, the forms of the two indicators are different. In other words, when cross-correlation is positive, the result shows how much lag there is between the two indicators, and when it is negative, the result shows how much lag there is between the change in *X(t)* and the opposing change in *Y(t)*. In this way, the signs of lag and cross-correlation for each combination of indicators were evaluated in this study. Moreover, significance in the negative region indicates that new PCR positive cases precede the other indicators, while significance in the positive region indicates that the other indicators precede new PCR positive cases.

Statistical significance was regarded as a two-sided P-value <0.05. All analyses were performed using R 3.6.3 (R Foundation for Statistical Computing, Vienna, Austria) and the Astsa package (version 1.10).

No approvals were needed as the study only used open-source data that did not identify any individuals.

## 3. Results

The timeline of changes in new PCR positive cases, the indicators of going out in public, and representative policies are shown in Fig 1.

For the first wave, the cross-correlation coefficients of new PCR positive cases and people flow were negative in the lag region –4 to 0 weeks (Fig 2A). For new PCR positive cases and the index of web searches for going outside, cross-correlation coefficients were significantly negative in the lag region –6 to –1 weeks (Fig 2B). For the number of times people browse restaurants, cross-correlation coefficients with new PCR positive cases were negative, with a lag between –4 to +1 weeks (Fig 2C). Finally, cross-correlation coefficients for the number of hotel guests and new PCR positive cases were negative between lag –3 to +1 weeks (Fig 2D) and were also positive in the lag region +10 to +12 weeks (Fig 2D). Therefore, the results show that new PCR positive cases preceded people flow and the index of web searches for going outside. Similarly, new PCR positive cases preceded or coincided with the number of times people

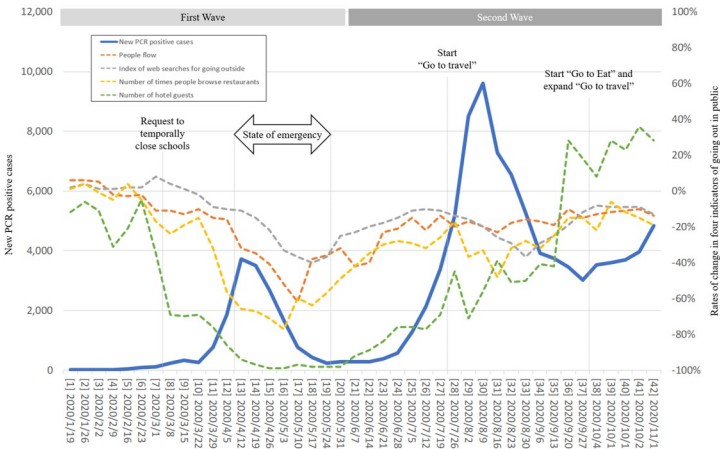

**Fig 1. Time trend of new PCR positive cases, four indicators related to going out in public, and representative policies.** The number before each date on the x-axis is the consecutive number of weeks, with Week 1 being January 19, 2020, the starting date of the data series.

browse restaurants and the number of hotel guests. Moreover, new PCR positive cases and the other four indicators had opposite forms since the cross-correlation coefficients were negative for all combinations of indicators.

Compared to the first wave, the second wave results were more complicated as the cross-correlation coefficients were negative, neutral (0), or positive, and the lag times were significant in the negative or positive regions, depending on the combination of indicators (Fig 3). The cross-correlation coefficients of new PCR positive cases and people flow were significantly negative in the lag region +8 to +9 weeks (Fig 3A), which means that the peak in new PCR positive cases lagged behind people flow (i.e., the peak in new PCR positive cases came after the people flow). This result was partially opposite to the first wave results, in which new PCR positive cases preceded people flow.

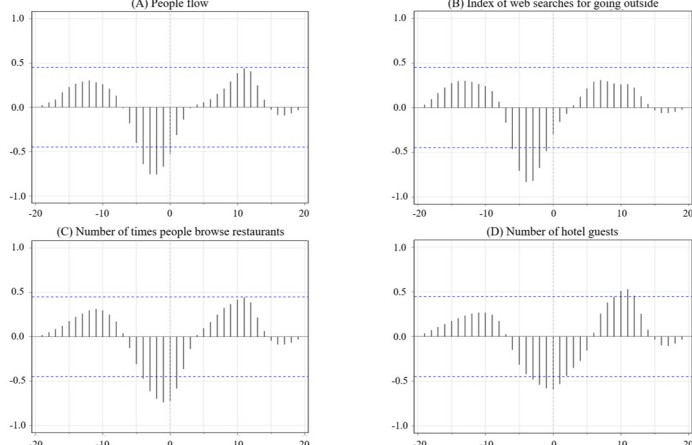

**Fig 2. Cross-correlation of new PCR positive cases with A) people flow, B) the index of web searches for going outside, C) the number of times people browse restaurants, and D) the number of hotel guests in the first wave.** The y-axis indicates the correlation coefficient and the x-axis indicates the lag. Dashed blue lines indicate 95% confidence intervals for a null model of no association.

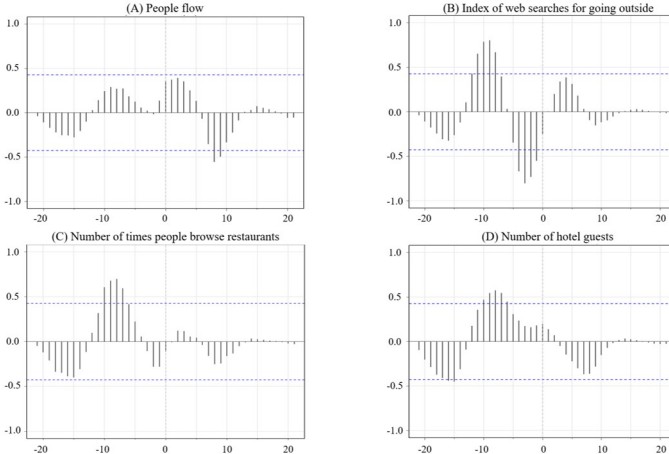

**Fig 3. Cross-correlation of new PCR positive cases with A) people flow, B) the index of web searches for going outside, C) the number of times people browse restaurants, and D) the number of hotel guests in the second wave.** The y-axis indicates the correlation coefficient and the x-axis indicates the lag. Dashed blue lines indicate 95% confidence intervals for a null model of no association.

Regarding the index of web searches for going outside, significant positive and negative cross-correlation coefficients with new PCR positive cases were found in distinct areas of the negative lag region (Fig 3B). Firstly, cross-correlation coefficients were significantly negative in the lag region –4 to –1 weeks (Fig 3B). This outcome confirms a similar structure to the first wave for this indicator combination. In other words, the peak of new PCR positive cases preceded the nadir of the index of web searches for going outside, and these two indicators were different. Secondly, cross-correlation coefficients were significantly positive in the lag region –12 to –8 weeks (Fig 3B). The peak (or nadir) in new PCR positive cases preceded the peak (or nadir) of the index of web searches for going outside, and the forms of the two indicators were similar. Such a structure was not observed in the first wave.

For the number of times people browse restaurants, cross-correlation coefficients with new PCR positive cases were significantly positive in the lag region –10 to –7 weeks (Fig 3C). Therefore, it showed that the peak (or nadir) of new PCR positive cases preceded the peak (or nadir) of the number of times people browse restaurants and that the two indicators were similar. This result partially contradicted the results from the first wave, as the two indicators differed in the first wave.

The number of hotel guests had significant positive and negative cross-correlation coefficients with new PCR positive cases in distinct negative lag regions. Firstly, the cross-correlation coefficients were significantly positive in the lag region –10 to –6 weeks (Fig 3D). The peak (or nadir) of new PCR positive cases preceded the peak (or nadir) of the number of hotel guests, and the forms of the two indicators were similar. This result partially contradicted the results of the first wave, as the forms of these two indicators were different in the first wave. Secondly, the cross-correlation coefficients were significantly negative in the lag region –16 to –15 weeks (Fig 3D). The peak in new PCR positive cases preceded the nadir of the number of hotel guests, and these two indicators were different.

## 4. Discussion

In this study, we analyzed the change in new PCR positive cases with four indicators concerned with people's behavior in terms of going out in public during the first and second waves of COVID-19 infection in Japan.

In the first wave, all four indicators were significantly associated in the negative lag region at or near the same time as the change in new PCR positive cases (Fig 2A–2D). This is consistent with previous research that showed the association between an increase in the number of new PCR positive cases and the decrease in people's mobility [45–50]. Additionally, this is also consistent with the fact that the declaration led people to stay home [51]. People followed the government's request to voluntarily refrain from going out in order to protect themselves from the virus by limiting their contact with others. Moreover, new information about COVID-19 continued to be disseminated daily, along with reports of new PCR positive cases. It is assumed that this information gave people a sense of urgency and encouraged them to refrain from going out [42, 43]. This behavior is likely attributable to increased risk perception in the early stages of the epidemic, and when self-restraint in terms of staying home became widely practiced [35–42].

The indicators of going out in public had various lag times in relation to new PCR positive cases. The starting point and the peak in significant lag time for the number of times people browse restaurants and the number of hotel guests were relatively earlier than the people flow and the index of web searches for going outside. This observation suggests that people were particularly aware of the risk of eating out and staying at hotels, and therefore refrained from these activities. This outcome is supported by the fact that restaurants, bars, and tourism activities were considered non-essential and were avoided towards the beginning of the epidemic [65, 66].

Compared to the first wave, three different characteristics were observed in the second wave. Firstly, the four indicators were mainly in an upward trend in the second wave, while in the first wave they trended downwards (Fig 1).

Secondly, a different association was observed for people flow, the number of hotel guests, and the number of times people browse restaurants at or near the same time as the change in new PCR positive cases. There were three possible reasons why the second wave's results differed from the results of the first wave. Firstly, the policies offered by the government influenced people's behavior; a state of emergency was not declared and the messages from the government in the second wave were not as strong or directive in comparison with its communication during the first wave. The results are consistent with previous research showing that different messages have different effects [37, 51, 67, 68]. Secondly, the "Go to Travel" and "Go to Eat" campaigns that encouraged people to go out may have influenced people's behavior. Although not statistically significant, the number of hotel guests showed a positive trend immediately after the change in new PCR positives cases, while in the first wave it showed a statistically significant negative trend. In addition, the negative lag in the number of times people browse restaurants was not statistically significant, while the lag was statistically significant in the first wave. These observations suggest that the "Go to Travel" and "Go to Eat" campaigns effectively promoted travel and eating out. This result is supported by previous research that shows an increase in the desire to go out, especially for leisure-related activities [69]. The difference between the number of times people browse restaurants and the number of hotel guests might be because the "Go To Eat" campaign started in October, while the "Go To Travel" campaign started on July 22. The timing of the change in the number of times people browse restaurants was late compared with the number of hotel guests [31–34]. Thirdly, people's experiences in the first wave, the "new lifestyle" advocated by the government at the end of the first wave, and the two factors mentioned above provided people with the sense that the risk of spreading COVID-19 would decrease soon, even when the infection started spreading again. This is consistent with studies in various countries showing that the effectiveness of lockdowns and the degree of compliance with restrictions diminish with the passage of time [52–54]. This result was likely attributable to the previous observation that the levels of

people's anxiety and their preventive behaviors decrease as their perception of the seriousness and immediacy of the threat reduced [44, 70]. The factors that encouraged people to stay at home during the first wave [35–42] have been recognized to some extent, and suggest that the sense of a crisis was not perceived beyond the first wave. In the second wave, the only change similar to that in the first wave was the index of web searches for going outside. However, the change in the index after the change in new PCR positive cases appeared faster and ended earlier than in the first wave. This finding may have resulted from people knowing how to coexist with the virus more effectively than during the first wave. Besides, while the index of web searches for going outside was decreasing, the people flow was not significantly reduced, suggesting that people were gradually resuming their public movements that did not require web searches such as commuting and their daily travel routines.

Finally, six to twelve weeks after the new PCR positive cases fluctuated, a significant positive change was seen in the number of hotel guests, the index of web searches for going outside, and the number of times people browse restaurants. Three reasons can be attributed to the observations. Firstly, the "Go To Travel" and "Go to Eat" campaigns were expanded six to twelve weeks after the rapid increase in new PCR positive cases. Secondly, almost all positive cases had recovered, and new PCR positive cases were nearing their nadir at the same time. Thirdly, people might have simply become complacent about COVID-19.

Comparing among four indicators of mobility in the second wave, the index of web searches for going out, and the number of times people browse restaurants showed a similar trend. Except for during zero to five weeks after the new PCR positive cases, people flow also fluctuated. This suggests that while people changed the frequency of their searching and browsing activities, there was no significant change in people flow despite the change in new PCR positive cases. The number of hotel guests formed differently, potentially due to the "Go To Travel" campaign.

## 5. Strength and limitations

Although some studies show the relationship between the degree of infection spread and people's behavior in terms of going out in public, only a few studies describe the relationship by showing the lags between the number of new PCR positive cases and going out behavior in Japan. The strength of this study is its demonstration of lags in the early stages of a pandemic as a baseline, as well as its identification of the differences in people's mobility and behavior between the first and second waves. Moreover, it is unique in showing the difference among the four indicators.

The data were sourced from the V-RESAS website. Since the data was gathered from private entities, it cannot include all people. However, the data used in this study is also used by policymakers and the media, so it is considered to have a level of validity [58]. There might be seasonal variations in people's behavioral changes. Since the analysis period was from January 16 to the first week of November, the results might have been underestimated or overestimated because of seasonal variations in behavior that might not have been considered in the analysis. However, since people's behavior during a pandemic differed significantly from a usual year, this limitation is not considered a serious problem.

## 6. Conclusions

The main findings of this study were as follows. Firstly, people refrained from going out in public during the first wave, but they did not refrain from going out in the second wave, even though there were more new PCR positive cases than in the first wave. Also, explicit differences among the four indicators were observed in the second wave compared to the first wave.

Moreover, results suggest that people going out in public in this context seems to relate to the policies and campaigns communicated to them. In conclusion, policies rather than new PCR positive cases might be more influential in affecting people's mobility. In the case of a possible future spread of the disease, other factors that influence public awareness should be considered.

## Supporting information

**S1 Dataset. Study source dataset.**
(XLSX)

## Author Contributions

**Conceptualization:** Hiromichi Takahashi, Iori Terada, Takuya Higuchi, Daisuke Takada, Jung-ho Shin, Susumu Kunisawa, Yuichi Imanaka.

**Data curation:** Hiromichi Takahashi, Iori Terada, Susumu Kunisawa.

**Formal analysis:** Hiromichi Takahashi, Iori Terada, Takuya Higuchi.

**Funding acquisition:** Yuichi Imanaka.

**Investigation:** Hiromichi Takahashi, Iori Terada, Takuya Higuchi, Daisuke Takada, Jung-ho Shin, Yuichi Imanaka.

**Methodology:** Hiromichi Takahashi, Iori Terada, Takuya Higuchi, Daisuke Takada, Jung-ho Shin.

**Project administration:** Yuichi Imanaka.

**Resources:** Hiromichi Takahashi, Iori Terada, Susumu Kunisawa, Yuichi Imanaka.

**Software:** Hiromichi Takahashi, Iori Terada, Takuya Higuchi, Daisuke Takada, Jung-ho Shin.

**Supervision:** Yuichi Imanaka.

**Validation:** Hiromichi Takahashi, Iori Terada, Takuya Higuchi, Daisuke Takada, Jung-ho Shin, Susumu Kunisawa, Yuichi Imanaka.

**Visualization:** Hiromichi Takahashi, Iori Terada.

**Writing – original draft:** Hiromichi Takahashi, Iori Terada.

**Writing – review & editing:** Hiromichi Takahashi, Iori Terada, Takuya Higuchi, Daisuke Takada, Jung-ho Shin, Susumu Kunisawa, Yuichi Imanaka.

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
