## [Decision Letter · Decision Letter 0]

23 Nov 2021

PONE-D-21-16674

The relationship between new PCR positive cases and going out in public during the COVID-19 epidemic in Japan

PLOS ONE

Dear Dr. Imanaka,

Thank you for submitting your manuscript to PLOS ONE. After careful consideration, we feel that it has merit but does not fully meet PLOS ONE’s publication criteria as it currently stands. Therefore, we invite you to submit a revised version of the manuscript that addresses the points raised during the review process.

Please see the full reviewer reports below. The reviewers have requested further information in the framing of the study, as well as ways to strengthen the conclusions. Please respond to the reviewer comments in full, and provide a marked up copy of the changes upon resubmission. Please also ensure that the manuscript is thoroughly copyedited, and that all data sources used are listed in the Data availability statement in the submission form.

We look forward to receiving your revised manuscript.

Kind regards,

Hanna Landenmark

Senior Editor

PLOS ONE

Journal Requirements:

"This work was supported by JSPS KAKENHI (Grant Number JP19H01075) from the Japan Society for the Promotion of Science (https://www.jsps.go.jp/english/e-grants/), and by the GAP Fund Program of Kyoto University, GAP Fund Program Type B (http://www.venture.saci.kyoto-u.ac.jp/?page_id=83#gp) to Y. I. The funders played no role in the study design, data collection and, data management, analysis, decision to publish, preparation, review and approval of the manuscript." 

We note that one or more of the authors is affiliated with the funding organization, indicating the funder may have had some role in the design, data collection, analysis or preparation of your manuscript for publication; in other words, the funder played an indirect role through the participation of the co-authors. If the funding organization did not play a role in the study design, data collection and analysis, decision to publish, or preparation of the manuscript and only provided financial support in the form of authors' salaries and/or research materials, please do the following:

a. Review your statements relating to the author contributions, and ensure you have specifically and accurately indicated the role(s) that these authors had in your study. These amendments should be made in the online form.

b. Confirm in your cover letter that you agree with the following statement, and we will change the online submission form on your behalf: 

“The funder provided support in the form of salaries for authors [insert relevant initials], but did not have any additional role in the study design, data collection and analysis, decision to publish, or preparation of the manuscript. The specific roles of these authors are articulated in the ‘author contributions’ section.

Reviewers' comments:

Reviewer's Responses to Questions

**Comments to the Author**

1. Is the manuscript technically sound, and do the data support the conclusions?

Reviewer #1: Yes

Reviewer #2: Partly

2. Has the statistical analysis been performed appropriately and rigorously? 

Reviewer #1: I Don't Know

Reviewer #2: Yes

3. Have the authors made all data underlying the findings in their manuscript fully available?

Reviewer #1: Yes

Reviewer #2: No

4. Is the manuscript presented in an intelligible fashion and written in standard English?

Reviewer #1: Yes

Reviewer #2: Yes

5. Review Comments to the Author

Reviewer #1: The authors performed the cross-correlation analysis of the relationship between indicators of people’s going out behaviors and the new PCR positive cases during the first and second waves of COVID-19 in Japan. Data collection was based on the online sources, including the Toyo Keizai Online and V-RESAS website report. Four indicators of people’s going out activities included the people flow (Agoop’s location data), the index of web searches for going outside (Yahoo search), the numbers of times people browse restaurants (Retty’s food data platform), and the numbers of hotel guests (Tourism Forecast platform).

The manuscript presented the interesting points aiming to connect the public awareness, the public health policy, the economic stimulus measure, and the new positive COVID-19 cases in Japan. However, these are indirect measures, which required careful interpretation. Some issues should be revised to strengthen the manuscript.

- The definition of the first wave in line 45 was defined as March-May 2020, but it was defined as January-May 2020 in lines 142-143.

- Line 72: Please clarify the characteristics of Japanese culture. Greeting culture?

- Redundant points in lines 74-75 and 89-90.

- Redundant points in lines 77-78 and 90-91.

- Check the pattern of the cited reference 46.

Available from:/pmc/articles/PMC4982160/?report=abstract)

- “As for the positive lag between new PCR positive cases and the number of hotel guests (lag +10 to +12; Figure 2D), this result is likely due to the data in the period before the

COVID-19 epidemic occurred.” Is this interpretation confirmed by the analysis that excluded the data in the period before the COVID-19 epidemic?

- Does Retty’s food data platform include the data on food delivery service? If it does, the increase in ordering food delivery might be the result of the decrease people flow. Please clarify this part.

- The discussion part should include the other studies on the people’s going out behavior during the COVID-19 outbreak in the other geographical areas. More references from the original articles should be added.

Reviewer #2: The authors studied the correlation between PCR positive cases and four indicators to going out in public in two waves in Japan. The methodology and the conclusion are clear and straightforward. I have several concerns that may need editor’s attention for the final decision.

Major concerns:

1. I would like to urge the authors to elaborate on the contributions of the study. The introduction tells the context of the research. However, to what extent the study will contribute to the existing literature is barely discussed. It seems that the authors did not conduct comprehensive literature review especially studies regarding the COVID-19. I am pretty sure that many studies have investigated the indicators to COVID-19 and its related non-pharmaceutical interventions. Many studies have investigated the correlation and lagged correlation as well.

2. For the social media data, representativeness is always a big concern. Whether the data could support the conclusion which works on the whole Japanese population while the data may be a small sample of the population. Also, the authors do not introduce how many data were collected for four indicators, which may be a big problem.

3. Correlation is not causality. The authors need to pay much attention to interpret the correlation results. Significant correlation does not mean that positive PCR led to the results of four investigated indicators.

Minor comments:

Page 8, line 126, I would like to invite authors to elaborate the choice of Yahoo Japan. Is it popular in Japan? Usually how many users? I am not familiar with the situation in Japan, but obviously in America, Google has much more users than Yahoo.

Page 10, ‘E[] denotes the mean’ loses the generic. E[] would better be expectation here.

Page 11, line 176. Please pay much attention to the input in R because the formula in R is different from the equation. If you input ccf(x, y) in R, then it is E[X(t+l)Y(t)] (only write the numerator here). Please double check.

6. PLOS authors have the option to publish the peer review history of their article (what does this mean?). If published, this will include your full peer review and any attached files.

Reviewer #1: No

Reviewer #2: No

---

## [Author Response · Author response to Decision Letter 0]

30 Jan 2022

Dear Editors and Reviewers,

Thank you very much for reviewing our manuscript and offering your valuable advice. We have

addressed your comments with point-by-point responses and revised the manuscript accordingly.

Responses to the Comments

Reviewer #1:

1. The definition of the first wave in line 45 was defined as March-May 2020, but it was defined as January-May 2020 in lines 142-143.

Response

Thank for your comment. We have amended the explanation in the introduction section.

Revise manuscript:

The first wave of COVID-19 infections ended in May 2020 (4).

(page 4, line 48 of the revised manuscript).

2. Line 72: Please clarify the characteristics of Japanese culture. Greeting culture?

Response

Thank you for your suggestion. We have added an explanation about the greeting characteristics to clarify this.

Revise manuscript:

In addition, the characteristics of Japanese culture and customs— greeting without shaking hands, hugging or kissing, and wearing cloth or paper facemasks to prevent respiratory infections and pollen allergies— may have contributed to the lower number of new polymerase chain reaction (PCR) positive cases and deaths per population compared to other countries (41).

(page 5, 6 line 75-79 of the revised manuscript).

3. Redundant points in lines 74-75 and 89-90.

Response

Thank you for pointing this out, we have deleted lines 89–90.

4. Redundant points in lines 77-78 and 90-91.

Response

Thank you for highlighting this; we have deleted lines 77–78.

5. Check the pattern of the cited reference 46.

Available from:/pmc/articles/PMC4982160/?report=abstract)

Response

Thank you for bringing our attention to this. We have changed the reference pattern.

Revise manuscript:

(page 33, line 558 of the revised manuscript. The new reference number is 62).

6. “As for the positive lag between new PCR positive cases and the number of hotel guests (lag +10 to +12; Figure 2D), this result is likely due to the data in the period before the COVID-19 epidemic occurred.” Is this interpretation confirmed by the analysis that excluded the data in the period before the COVID-19 epidemic?

Response

Thank you for your question. We did not confirm this interpretation by the analysis that excluded the data in the period before the COVID-19 epidemic. However, we believe that the long lag (+10 to +12) has no significant meaning, given people did not determine their actions based on an assessment of the previous two or three months. As such, we have deleted this sentence.

Revise manuscript:

(page 18, line 279 of the revised manuscript).

7. Does Retty’s food data platform include the data on food delivery service? If it does, the increase in ordering food delivery might be the result of the decrease people flow. Please clarify this part.

Response

Thank you for your question. The Retty’s site includes information on restaurants, where people can search for restaurants, select their favorites, and make an online reservation. Although this site shows whether a restaurant offers a delivery service, people cannot directly order food delivery from Retty’s. Instead, many people use delivery services such as Uber or the homepage of individual restaurants. Therefore, it seems that a few people wanting food delivery visit this site.

Revise manuscript:

No revise.

8. The discussion part should include the other studies on the people’s going out behavior during the COVID-19 outbreak in the other geographical areas. More references from the original articles should be added.

Response

Thank you for your suggestion. We have reviewed and added additional studies in the discussion. References 45–54 and 65–70 are new additions to the introduction and discussion sections.

Revise manuscript:

page 6, line 90-93, page 17, line 263, 264, page 18, line 279, 291, page 19, line 300, 309 of the revised manuscript.

Reviewer #2:

Major concerns:

1. would like to urge the authors to elaborate on the contributions of the study. The introduction tells the context of the research. However, to what extent the study will contribute to the existing literature is barely discussed. It seems that the authors did not conduct comprehensive literature review especially studies regarding the COVID-19. I am pretty sure that many studies have investigated the indicators to COVID-19 and its related non-pharmaceutical interventions. Many studies have investigated the correlation and lagged correlation as well.

Response

Thank you for your suggestion. We have reviewed and added further literature to the introduction and discussion section. Moreover, we have addressed and clarified the contribution of this study in the introduction, as well as in the strength and limitations section.

Revise manuscript:

Previous studies showed the association between an increase in the number of new PCR positive cases and a decrease in people’s mobility (45–50), and that the government declaration is effective to lead people to stay home (51). In addition, studies in various countries showed that the degree of compliance with restrictions diminished over time (52–54).

 The association between the new PCR positive cases and going out in public, the differences in people going out between the first and subsequent waves, and people’s behavior among the different kinds of going out have not been clarified, especially in Japan. Thus, this study investigated the relationship between the new PCR positive cases and four indicators of going out, namely the people flow; the index of web searches for going outside; the number of times people browse restaurants, and the number of hotel guests, by examining the lag of going out behaviors from the spread of infection. By examining the changes in people going out in public between the first and subsequent waves, and the underlying changes in people's awareness of the crisis as well as the effects of government policies, this study will provide insight for implementing effective countermeasures against the spread of infection.

(page 6, line 94-103 of the revised manuscript)

Although some studies show the relationship between the degree of infection spread and people’s behavior in terms of going out in public, only a few studies describe the relationship by showing the lags between the number of new PCR positive cases and going out behavior in Japan. The strength of this study is its demonstration of lags in the early stages of a pandemic as a baseline, as well as its identification of the differences in people’s mobility and behavior between the first and second waves. Moreover, it is unique in showing the difference among the four indicators.

(page 21, line 338-344 of the revised manuscript)

2. For the social media data, representativeness is always a big concern. Whether the data could support the conclusion which works on the whole Japanese population while the data may be a small sample of the population. Also, the authors do not introduce how many data were collected for four indicators, which may be a big problem.

Response

Thank you for your comment. All four sites where the indicators were collected are famous and contain large amounts of data. Below is additional information about the four sites:

• Agoop

According to the document issued by Ministry of Economy, Trade and Industry, The number of Agoop's smartphone app users is enlarged to the total population of Japan. As of 2019, the number of app users was 1 million annually. https://www.kanto.meti.go.jp/seisaku/kikaku/data/bunseki_shuho_v3.pdf

• Yahoo! Japan

According to Statcounter, Google’s search engine market share in Japan is around 75%, compared to Yahoo’s 20%, as of November 2021. On the other hand, Google’s market share worldwide is more than 90% while Yahoo’s is less than 2%. 

https://gs.statcounter.com/search-engine-market-share

Research by Nielsen, however, shows that Yahoo is the most popular site in Japan:

Access from Desktop:

Yahoo! Japan – 33.7 million / month

Google – 23.9 million / month

Access from Smartphone:

Google – 60.5 million / month

Yahoo! Japan – 56.3 million / month

This data indicates that Yahoo is more popular than Google in Japan. Yahoo is the most visited site in Japan, while Google is the most used search engine in Japan. Therefore, Yahoo is used widely and is considered one of the most popular sites in Japan.

• Retty

Retty is one of the biggest restaurant portal websites nationwide. Retty was accessed by approximately 70 million people and viewed more than 200 million times during one month, according to Similarweb. Among the five largest restaurant portal sites, the number of people who accessed and viewed the Retty pages were the second and third biggest, respectively (https://hirakudayo.org/hikaku5/). In addition, ‘TOYOKEIZAI’, one of the most famous magazine publishers in Japan, noted that Retty is one of the four best sites in Japan. https://toyokeizai.net/articles/-/144841

• Tourism Forecast Platform

The Tourism Forecast Platform collected the data of 200 million staying in Japan, compared with the total estimated number of 560 million. https://kankouyohou.com/docs/kankouyohou_info.pdf

Revise manuscript:

Although there are no revise in the manuscript, we can write additional information if needed.

3. Correlation is not causality. The authors need to pay much attention to interpret the correlation results. Significant correlation does not mean that positive PCR led to the results of four investigated indicators.

Response

Thank you for your comment. We have stopped using words that might have suggested causality, such as “react”, “reaction” and “respond.”

Minor comments:

4. Page 8, line 126, I would like to invite authors to elaborate the choice of Yahoo Japan. Is it popular in Japan? Usually how many users? I am not familiar with the situation in Japan, but obviously in America, Google has much more users than Yahoo.

Response

Thank you for your suggestion. Please see major comment 2 for further background and selection rationale for Yahoo! Japan.

Revise manuscript:

No revise.

5. Page 10, ‘E[] denotes the mean’ loses the generic. E[] would better be expectation here.

Response

Thank you for your comment. As advised, we have changed the expression from “mean” to “expectation”.

Revise manuscript:

(page 10, line 161 of the revised manuscript).

6. Page 11, line 176. Please pay much attention to the input in R because the formula in R is different from the equation. If you input ccf(x, y) in R, then it is E[X(t+l)Y(t)] (only write the numerator here). Please double check.

Response

Thank you for your advice. We had already considered that the definition of the CCF used in this study differs from that of the function used in the R package. As such, we reversed the two inputted time series data, X and Y.

Revise manuscript:

No revise.

Again, thank you for allowing us to revise our manuscript and incorporate your valuable comments and insights.

---

## [Decision Letter · Decision Letter 1]

21 Mar 2022

The relationship between new PCR positive cases and going out in public during the COVID-19 epidemic in Japan

PONE-D-21-16674R1

Dear Dr. Imanaka

We’re pleased to inform you that your manuscript has been judged scientifically suitable for publication and will be formally accepted for publication once it meets all outstanding technical requirements. Within one week, you’ll receive an e-mail detailing the required amendments. When these have been addressed, you’ll receive a formal acceptance letter and your manuscript will be scheduled for publication.

Kind regards,

Edris Hasanpoor

Academic Editor

PLOS ONE

---

## [Editor Report · Acceptance letter]

24 Mar 2022

PONE-D-21-16674R1 

The relationship between new PCR positive cases and going out in public during the COVID-19 epidemic in Japan 

Dear Dr. Imanaka:

I'm pleased to inform you that your manuscript has been deemed suitable for publication in PLOS ONE. Congratulations! Your manuscript is now with our production department. 

Kind regards, 

on behalf of

Dr. Edris Hasanpoor 

Academic Editor

PLOS ONE